

# IS MASS TRANSFER IN SECONDARY ORGANIC AEROSOL PARTICLES INTRINSICALLY SLOW? EQUILIBRATION TIMESCALES OF ENGINE EXHAUST AND A-PINENE SOA UNDER DRY AND HUMID CONDITIONS

Khairallah Atwi[1,2], Mohamad Baassiri[1], Mariam Fawaz[1,3], Alan Shihadeh[1]

[1]Aerosol Research Laboratory, Department of Mechanical Engineering, American University of Beirut, Beirut, Lebanon
[2]Current address: Air Quality and Climate Research Lab, University of Georgia, Athens, Georgia, USA
[3]Current address: University of Illinois at Urbana-Champaign, Urbana, Illinois, USA

*Correspondence to*: Alan Shihadeh (as20@aub.edu.lb)

**Abstract.** Semi-volatile secondary organic aerosols (SOA) comprise a major fraction of ambient particle pollutants. The partitioning of SOA in the atmosphere has commonly been assumed to be fast enough that it could be computed solely from thermodynamic equilibrium considerations e.g., using Raoult's Law. This simplifying assumption has been called into question by recent studies of single SOA particles evaporating in a zero-vapor concentration environment, which reported unexpectedly slow evaporation relative to atmospheric timescales. In this work we directly investigated the phase equilibration

kinetics of systems of SOA particles under realistic atmospheric conditions. SOA was generated in an oxidation flow reactor (OFR) from engine exhaust or α-pinene and mixed with clean air in an atmospheric pressure smog chamber (32° C) to induce evaporation. The evolution of the particle size distribution was monitored over time as the aerosol system returned to phase equilibrium under different particle concentrations (2.5 and 5 µgm$^{-3}$) and humidity conditions (<10% and 60%). We found that under typical ambient conditions, and independent of relative humidity and precursor origin (engine exhaust vs. α-pinene),

SOA reestablished equilibrium with the vapor phase within minutes, and that the evolution of particle size was well-fit by a computational model treating the particle phase as well-mixed. The effective thermodynamic saturation concentration of the SOA was found to be in the range 0.02-0.11 µgm-3 at 20 °C, assuming an enthalpy of vaporization of 150 kJmol$^{-1}$. The effective evaporation coefficient was found to be in the range 0.1-0.2 using a gas diffusion coefficient of $5\times10^{-6}$ m$^2$s$^{-1}$. Unlike previous single-particle studies, this data suggests that under most loading conditions, anthropogenic and biogenic SOA can

rapidly attain phase equilibrium in the atmosphere and that their partitioning can be modeled assuming thermodynamic equilibrium.

## 1 Introduction

Organic aerosols (OA) have a substantial impact on climate. In the atmosphere, they act as cloud condensation nuclei, changing the properties of clouds and causing significant changes in precipitation patterns (Boucher et al., 2013). They scatter and absorb

solar and longwave radiation and can alter the earth's radiative budget (Myhre et al., 2013). OA are also harmful to human



health, with prolonged exposure to high concentrations linked with lung cancer and cardiopulmonary diseases (Pope III et al., 2002;Davidson et al., 2005). A large fraction of atmospheric OA is formed through photochemical conversion of gaseous precursor species in the atmosphere (Kroll and Seinfeld, 2008). Formulating effective policies to limit ambient OA therefore requires understanding the formation mechanisms and aging of SOA. However, despite concerted effort over the past decade,

chemical models still have trouble reproducing observed concentrations of SOA, particularly around urban areas (Heald et al., 2005;Spracklen et al., 2011;Tsigaridis et al., 2014;Hodzic et al., 2016;Koffi et al., 2016).

Organic compounds partition between the gas and particle phases depending on their volatility and the total concentration of condensed organics (Pankow, 1994). As volatile organic compounds react with hydroxyl radicals, ozone, or other oxidants, their volatility decreases, leading to the formation of SOA (Kroll and Seinfeld, 2008). In the atmosphere, chemical reactions

in the gas or condensed phases further change the volatility of these compounds and thus drive them to repartition accordingly. In addition, temperature fluctuations or dilution can perturb the thermodynamic equilibrium, causing the particles to evaporate or condense.

The complexity of the processes OA undergo in the atmosphere makes their explicit representation in chemical transport models (CTMs) highly demanding. Consequently, organic compounds are often lumped together in models according to

properties such as volatility, extent of oxidation, or polarity (Odum et al., 1996;Donahue et al., 2006;Donahue et al., 2011;Pankow and Barsanti, 2009). The effects of chemical aging and other processes are modeled as changes to these properties. Most models then assume that the consequent change in gas-particle partitioning is immediate, in that it occurs on timescales shorter than those of the time steps used in those models (Gaydos et al., 2007;Robinson et al., 2007;Hodzic et al., 2010). The actual kinetics of the processes of condensation and evaporation, however, involve several steps that could be rate

limiting. Slow species diffusion within particles, for example, could significantly retard equilibration, as the diffusion of the evaporating species to the particle surface is inhibited. Interfacial phenomena (e.g. molecular accommodation or evaporation coefficient far below unity) can also be rate-limiting.

Several groups have reported that ambient and lab-generated SOA can exist in a highly viscous state, with correspondingly low rates of intraparticle species diffusion (Virtanen et al., 2010;Saukko et al., 2012;Perraud et al., 2012). The formation of a

shell layer of secondary organic compounds on particle surfaces has been reported to retard the diffusion of molecules to the surfaces or the uptake of molecules from the surrounding gas (Zhou et al., 2013;Abramson et al., 2013). For example, in controlled laboratory experiments with SOA particles having a core and shell morphology, Loza et al. (2013) found that the extent of evaporation decreased when the shell consisted of SOA, suggesting that evaporation of the more volatile compounds in the particle core was hindered by slow transport through the low volatility SOA shell. Similarly, in a series of thermodenuder

experiments, Cappa and Wilson (2011) studied evaporation of lubricating oil droplets and α-pinene SOA. They found that, in contrast to lubricating oil, the chemical composition of α-pinene SOA did not change significantly as particles evaporated, suggesting that the more volatile components in the particle core were unable to reach the surface. Furthermore, they found that an evaporation coefficient on the order of $10^{-4}$ was needed to fit their measurements to an evaporation model that neglected intraparticle diffusion limitations and that used a volatility distribution from a previous study. Other groups studying SOA



particle evaporation have also reported that model predictions made assuming a single well-mixed particle phase (i.e. one in which mass transport within the particle was infinitely fast) required unrealistically small evaporation coefficients in order to match observed particle evaporation rates (Vaden et al., 2011;Stanier et al., 2007;Grieshop et al., 2007). In these studies, slow evaporation of both ambient and lab-generated SOA and their disagreement with model predictions were attributed to limited

intra-particle diffusion.

Interestingly, another body of literature is emerging that counters the notion that SOA particle phase change is intrinsically limited by slow intraparticle mass transfer. In experimental observations of evaporation of particles consisting of α-pinene SOA and squalene, Robinson et al. (2015) found that even under very dry conditions (i.e. RH = 5%), whether squalene coated the SOA or was coated by SOA did not alter the extent or timescales of evaporation of the more volatile squalene, and

concluded that the presence of an α-pinene SOA shell did *not* inhibit mass transfer. In a similar vein, in controlled laboratory studies of evaporation of SOA produced from ozonolysis of α-pinene, Saha and Grieshop (2016) and Saleh et al. (2013b) reported evaporation coefficients on the order of 0.1. Similarly, in studies of ambient SOA, Saleh et al. (2012) and Saha et al. (2017) reported effective evaporation coefficients in the range of 0.2-0.5, several orders of magnitude greater than reported by the aforementioned literature.

Further complicating the picture is a recent study by Ye et al. (2016) which found intraparticle mass transfer resistance in SOA was a function of relative humidity; resistance increased when relative humidity was below 40%. Gorkowski et al. (2017) showed that, at 75% RH, a shell of α-pinene SOA did not inhibit the mass transfer of a core formed of water, glycerol, or squalane.

It is noteworthy that in studies where SOA observations were fitted to models to infer whether the evaporation was "slow",

the investigators normally relied on values of vapor pressure reported in previous smog chamber studies as inputs to the models. If these assumed values happened to be higher than those of the SOA under observation, then "slow" evaporation might have simply reflected the fact that the SOA under study had a lower volatility than assumed, rather than reflecting intrinsic mass transfer limitations. That is, evaporation can appear "slow" if the volatility is overestimated and/or if mass transfer is inhibited (e.g., as reflected by a low effective evaporation coefficient). It has been reported that atmospheric OA is largely composed

of compounds with low volatilities that do not match those estimated by traditional smog chamber experiments (Hallquist et al., 2009). In addition, aged laboratory-generated SOA may also contain non-volatile oligomers produced due to reactions in the condensed phase (Gao et al., 2004;Heaton et al., 2007;Tolocka et al., 2004). Taken together, these uncertainties suggest that caution is needed when interpreting evaporation data and highlight the difficulty of interpreting observations of evaporation kinetics when the methods of study do not simultaneously constrain the volatility. Importantly, all the studies that

simultaneously determined both the volatility and the effective evaporation coefficients of SOA particles (Saha and Grieshop, 2016;Saleh et al., 2013b;Saleh et al., 2012) reported evaporation coefficients much closer to unity than those that did not.

Another area of uncertainty derives from the exclusive focus of laboratory studies on SOA derived from biogenic precursors (e.g. α-pinene). Anthropogenic SOA is prevalent, particularly in urban areas (Gentner et al., 2017), yet has not been systematically examined in controlled laboratory experiments. In this work, we studied the evaporation kinetics of engine





exhaust SOA generated in an oxidation flow reactor (OFR) using a phase equilibration approach (Saleh et al., 2012) that simultaneously determines effective aerosol vapor pressure and evaporation coefficient. For comparison, we also examined α-pinene SOA. The experimental approach involved injecting a volume of SOA into a slightly heated chamber filled with clean air and then observing the evolution of the particle size distribution over several hours as the particles returned to phase equilibrium with the surrounding gases. The experiments were conducted at particle concentrations typically found in ambient air (several $\mu gm^{-3}$). The observations were fitted to a mathematical model of aerosol evaporation (Saleh et al., 2011) to determine effective vapor pressure and evaporation coefficient, as described below.

## 2 Theory

When a parcel of semi-volatile organic aerosol particles, initially in phase equilibrium with the surrounding gases, is subjected to a drop in the saturation ratio (*SR*) by heating or dilution, the particles respond by evaporating. If the initial particle loading is sufficient to re-saturate the gas phase, a new equilibrium condition will be attained at a lower particle mass concentration and smaller particles. The process by which the system *SR* returns to equilibrium is quasi-first order (Saleh et al., 2011), and is characterized by an equilibration timescale $\tau = 1/2\pi N_{tot}\, d_{p,in} DF$ where $N_{tot}$ is the total number concentration of the particles, $d_{p,in}$ is the initial diameter of the particles, D is the binary diffusion coefficient, and *F* is a correction factor for non-continuum effects. Importantly, $\tau$ is not a function of the vapor pressure of the evaporating species; high volatility aerosols evaporate more rapidly, but also require more total evaporation to attain a given change in *SR*. See Saleh et al. (2011) for further elaboration on this often confused subject.

While most variables in $\tau$ are well-constrained, the evaporation coefficient, $\alpha$, must usually be determined empirically. As in Saleh et al. (2009), the evaporation coefficient in this study is determined by fitting measured evaporation rates to a model that neglects intra-particle mass transfer resistance. Therefore, the empirically determined value is actually an effective evaporation coefficient, which represents the net effect of all mass transfer resistances including intraparticle diffusion and interfacial effects (e.g. mass accommodation).

We determine the effective evaporation coefficient of anthropogenic and biogenic SOA through isothermal dilution in a 5m³ Teflon chamber. In particular, SOA of initial mass concentration $C_0$ and with saturation ratio *SR=1*, is injected into a chamber filled with clean air, causing the SR to drop and evaporation to commence. The aerosol size distribution in the chamber is monitored continuously and fitted to an aerosol kinetic model. The fit determines the effective evaporation coefficient and saturation concentration that provide the minimum error between measured and computed particle sizes as the system returns to phase equilibrium. Importantly, this "phase equilibration method" does not require *a priori* knowledge about aerosol volatility (Saleh et al., 2012).

After injection into the chamber, the change in the particle size distribution is driven by two physical phenomena: particle evaporation and wall deposition. Because particle deposition on the walls of the chamber is simultaneous with evaporation, it





is likely that both suspended and deposited particles contribute vapor mass during phase equilibration and thus affect the rate at which phase equilibrium is approached. As in previous work (Stanier et al., 2007;Ranjan et al., 2012;Saleh et al., 2013a), it is assumed that particles deposited on the wall evaporate in accordance with the Maxwell equation (Maxwell, 1860), modified by a wall-bound contribution factor, $\omega$, whose value can vary between 0 and 1. At $\omega=0$, particles deposited on the walls are assumed not to evaporate, thus making no contribution to gas phase concentration in the chamber. At $\omega=1$, particles on the walls are assumed to act as if they were still suspended, and thus evaporate until equilibrium is reached. This factor is introduced into the Maxwell equation to account for the contribution of particles on the wall to vapor buildup in the chamber. Using this approach, it can be shown that, the vapor mass concentration, $C_g$, and particle diameter, $d_p$, are governed by Eqs. (1), (2), and (3) for a single component monodisperse aerosol:

$$\frac{dC_g}{dt} = 2\pi D d_p F\left(C_{sat} - C_g\right)\left(N_{sus} + \omega N_{wall}\right) \tag{1}$$

$$\frac{d(d_p)}{dt} = -\frac{4D}{\rho_p}\frac{F}{d_p}\left(C_{sat} - C_g\right) \tag{2}$$

$$\frac{dN_{sus}}{dt} = -\beta N_{sus} = -\frac{dN_{wall}}{dt} \tag{3}$$

where $C_{sat}$ is the vapor concentration at saturation, $N_{sus}$ is the number concentration of suspended particles, $N_{wall}$ is the number of particles deposited on the walls of the chamber per unit chamber volume, $\beta$ is the wall loss rate constant ($s^{-1}$), and $\rho_p$ is the aerosol density. For the non-continuum correction, the Fuchs-Sutugin (Fuchs and Sutugin, 1970) expression is used:

$$F = \frac{1 + Kn}{1 + \left(\frac{4}{3\alpha} + 0.377\right)Kn + \frac{4}{3\alpha}Kn^2} \tag{4}$$

where $Kn$ is the Knudsen number defined as $Kn = 2\lambda / d_p$ and $\alpha$ is the evaporation coefficient. In accordance with the derivation of Fuchs and Sutugin (1970), the mean free path is computed as $\lambda = 3D_{SOA,air} / \overline{c}$, with $\overline{c}$ representing the mean molecular speed. For a multicomponent, polydisperse aerosol with $n$ discrete size bins, the above equations can be written in terms of the effective thermodynamic and transport properties $C_{sat,eff}$, $\alpha_{eff}$ (Saleh et al., 2012), the condensation sink, and the condensation sink diameter. The condensation sink (Wexler and Seinfeld, 1990) is a measure of the molecular flux ($s^{-1}$) between all the particles and the surrounding vapor phase, and is given by:

$$CS = 2\pi D \sum_{i=1}^{n} F d_{p,i} N_i = 2\pi DFN d_{cs} \tag{5}$$

In Eq. (5), $d_{cs}$ is the condensation sink diameter, or the diameter of a monodisperse aerosol that would exhibit the same evaporation rate as a polydisperse aerosol with an equal number concentration.





Substituting Eq. (5) into Eq. (1) and replacing $d_p$ by $d_{cs}$ in Eq. (2) we obtain:

$$\frac{dC_g}{dt} = \left(CS_{sus} + \omega CS_{wall}\right)\left(C_{sat} - C_g\right) \tag{6}$$

$$\frac{d(d_{cs})}{dt} = \frac{-4D_{SOA,air}}{\rho_p}\frac{F}{d_{cs}}\left(C_{sat} - C_g\right) \tag{7}$$

Thus, equations (3), (6), and (7) describe the evaporation and change in diameter of a monodisperse aerosol trapped in a chamber as it approaches equilibrium with the surrounding gas. Most of the physical and transport properties in the above system of equations can be readily estimated or measured, leaving $C_{sat}$ and $\alpha$ as the unknowns. $C_{sat}$ is determined from the change in particle size between the initial and final equilibrium states of the aerosol. In particular, $C_{sat}$ is determined as the saturation concentration which, after equilibrium has been reached, will have brought about the observed reduction of the condensation sink diameter (i.e. $C_{sat}$ only depends on the initial and final size distributions, and not the rate at which the process proceeds). Once $C_{sat}$ has been determined, $\alpha$ is obtained by minimizing the error between $d_{cs}$ computed from Eq (7) and that of the particle distribution during the equilibration process. This approach is illustrated in Figure 1, which shows model predictions for the $d_{cs}$ at different values of $\alpha$ and $C_{sat}$. While the saturation concentration only determined the equilibrium value of $d_{cs}$, $\alpha$ dictates the rate at which this value is approached. Thus, for each experimental data set, there is a unique optimal combination of $\alpha$ and $C_{sat}$.

## 3 Experimental Approach

SOA is generated by feeding diluted engine exhaust (anthropogenic) or α-pinene (biogenic) to an oxidation flow reactor (OFR). The SOA is then injected into a heated Teflon™ chamber filled with clean air, causing the saturation ratio of the particles to drop suddenly to near zero, and evaporation to commence. The particle size distribution is monitored over several hours starting from the initial injection. Measurements are made under dry (<10% RH) and humid conditions (60% RH); humidity in the chamber and OFR are controlled and matched to either the dry or humid condition. The SOA generated under high humidity in the OFR is expected to have higher water content, leading to a less viscous phase state and more rapid intraparticle species diffusion (Grayson et al., 2016;Song et al., 2015). To the extent that particle evaporation is kinetically limited by high viscosity and slow intraparticle diffusion, the humid condition will reach equilibrium more rapidly, and be characterized by a larger evaporation coefficient.

Table 1 shows the conditions and the number of repeated measurements at each condition. A nominal baseline particle mass concentration of $C_{OA} = 5\mu gm^{-3}$ was selected to represent a typical urban atmospheric SOA concentration (Jimenez et al., 2009). We also conducted measurements at $C_{OA} = 2.5\mu gm^{-3}$ to verify that the observed arrest in particle evaporation at the baseline condition of $C_{OA} = 5\mu gm^{-3}$ was due to equilibration between particle and vapor phases, and not due to the depletion of volatile species from the particles. If the low concentration condition resulted in smaller final particle sizes than the high concentration



condition, then this was taken to indicate that the particles had indeed reached phase equilibrium rather than being depleted of volatile material.

## 3.1 Setup

### 3.1.1 Volatile precursors

Figure 2 shows the experimental setup for the experiments using diluted engine exhaust. A four-stroke single cylinder gasoline engine (Honda SHX1000, displacement: 49cc, compression ratio: 8.0:1, splash lubrication) was used as the source of anthropogenic emissions. The engine was operated at zero external load using 98 octane rating gasoline and SAE-40 motor oil.

The engine exhaust was sampled through a rotating disk dilutor (Testo Engineering, MD19-3E) operated at 800:1 dilution.
The line drawing the exhaust to the dilutor was heated to 120 °C to avoid the condensation of the vapor emissions. The dilutor head was also heated to 120 °C. 5 lpm of the diluted engine exhaust were directed to the OFR, where they were further diluted through mixing with a 20 lpm stream of zero air. This resulted in a total dilution of 3200:1, which is of a similar order of magnitude to that 10 minutes downstream of a highway (Zhang and Wexler, 2004).

To introduce evaporated α-pinene into the OFR, approximately 1 ml of liquid α-pinene was placed in a 15ml glass vial. Three
1 mm holes were drilled in the vial cap, allowing the diffusion of the evaporated α-pinene. The vial was placed inside a 250ml Büchner flask and 300 mlpm of dry air was continuously pushed through the flask and into the OFR.

### 3.1.2 Oxidation flow reactor

The OFR is a 64 L stainless steel rectangular chamber equipped with a mercury UV lamp (BHK Analamp, Model No. 82-9304-03) and a small fan to ensure its contents are well-mixed. The lamp emits radiation at both 185 and 254 nm, producing
both ozone and OH (Lambe et al., 2011). The OFR has three inlets, allowing independently controlled flows of dry air, humid air, and precursor-containing air. The temperature and humidity in the reactor are monitored using a probe (Vaisala HMP60). For all conditions, the temperature of the OFR was approximately 24 °C. In order to control reactor humidity, dry and humid compressed air streams were independently introduced into the OFR using two mass flow controllers (Omega FMA5500A). The humid flow was generated by pushing air through a heated Büchner flask partially filled with de-ionized water. Figure
S1 illustrates the typical temporal responses of ozone and SOA concentrations as a function of lamp operation and introduction of engine exhaust into the OFR.

SO$_2$ calibration measurements were conducted to estimate the equivalent extent of oxidation in the reactor as in (Kang et al., 2011). Assuming an atmospheric OH concentration of $1.5 \times 10^6$ molec.cm$^{-3}$ (Mao et al., 2009), the residence time in the reactor under dry conditions was equivalent to $6.3 \pm 0.9$ days of atmospheric aging.



### 3.1.3 Evaporation chamber

The SOA from the OFR was diluted in a 5 m$^3$ Teflon chamber made of 5 mil perflouoroalkoxy alkane (PFA) manufactured by Welch fluorocarbon. The chamber was housed inside a temperature-controlled enclosure maintained at $32 \pm 0.2$ °C. The environment inside the chamber was monitored using a temperature and humidity probe (Vaisala, Model HMP60). In both
dry and humid experiments, the humidity in the chamber matched that of the reactor. To verify that there was no thermal stratification inside the chamber, thermocouples were used to monitor the temperature at 3 different elevations at the middle of the chamber.

### 3.1.4 Particle sizing

A scanning mobility particle sizer (SMPS) (Classifier: TSI 3082, CPC: TSI 3772) was placed inside the temperature-controlled
enclosure and used to measure the particle size distribution in the Teflon chamber and the OFR. The SMPS was operated with a scan time of 60 seconds, a retrace of 4 seconds, and a purge of 20 seconds. The aerosol flow rate was restricted by the CPC to 1 lpm. The sheath flow was set to 4 lpm. Under these conditions, the SMPS measured particles in the range of 10-500 nm. In preliminary measurements, an aerodynamic particle sizer (TSI 3321), which measures particles in the 500nm to 20μm size range, was used to verify that the particles generated in the OFR were within the 10-500 nm measurement window of the
SMPS.

### 3.2 Procedure

Figure 3 shows the experimental procedure timeline. The chamber was flushed overnight with HEPA filtered dry or humid air, depending on the experimental conditions. Measurements of the particle size distribution began in the OFR prior to the introduction of any precursor. Once the precursors were added, several air changes were allowed to pass before the UV lamps
were turned on. This led to a rapid rise in particle number and mass, as shown in Figure 4.
Once a steady SOA distribution was obtained in the reactor, the air in the chamber was sampled to ensure it was particle free (< 10 particles/cm$^3$). The exhaust from the reactor was redirected into the chamber. The filling process continued until the mass concentration in the chamber reached the desired initial value (typically 12 minutes for the baseline condition of $C_{OA}$=5 μgm$^{-3}$). The particle size distribution was measured for approximately 300 minutes, a time found sufficient under all conditions
for the system to reach equilibrium.

### 3.3 Data analysis

### 3.3.1 Correction for wall effects on sink diameter

In all SOA experiments, we found that after an initial rapid drop in $d_{cs}$, the $d_{cs}$ began to grow gradually, eventually exceeding the original diameter. This is consistent with overlapping phenomena of particle evaporation and wall deposition. Because
small particles are more mobile than large particles, they are preferentially lost to the walls, causing the average particle size



of the suspended aerosol to increase in time, as has been well-reported in the literature (Crump et al., 1982;Okuyama et al., 1986). In all our observations, the size increase in time was highly linear following the initial evaporation phase. Thus, to correct for wall effects on $d_{cs}$ we used a first-order expression:

$$d_{cs,corrected} = d_{cs,measured} - kt \qquad (8)$$

where $k$ is the growth constant fitted to $d_{cs}$ after equilibrium has been reached.

To verify that the initial drop in size was due to evaporation, we also repeatedly injected into the chamber a non-volatile aerosol (dry ammonium sulfate) of the same size range as the SOA. As shown in Figure 5, only the SOA exhibited the initial drop in $d_{cs}$. As with SOA, the growth in $d_{cs}$ was linear in time.

### 3.3.2 Determining $\alpha$ and $C_{sat}$

The evaporation model described in section 2 was coded in MATLAB using a variable time-step stiff equation solver to produce time series predictions of $d_{cs}$ for various values of $C_{sat}$ and $\alpha$. Consistent with Saleh et al. (2013b), the density, molar mass, and gas-phase diffusion coefficient for SOA were assumed to be $\rho = 1250 \, \text{kgm}^{-3}$, $M = 0.2 \, \text{kgmol}^{-1}$, and $D = 5 \times 10^{-6} \, \text{m}^2\text{s}^{-1}$. Since the final extent of evaporation is independent of the rate of evaporation, the saturation concentration could be calculated using any assumed value of $\alpha$. Once the $C_{sat}$, which minimized the error between the predicted and measured steady-state $d_{cs}$

was determined, simulations were generated with various values of $\alpha$ in steps of 0.01, and the $\alpha$ which produced the minimum least squares error between the predicted and measured time series of $d_{cs}$ during evaporation was identified.

A Monte Carlo approach was employed to estimate uncertainty in $\alpha$ and $C_{sat}$ arising from measurement uncertainty in particle size, count, molar mass, and the gas diffusion coefficient, as described in (Saleh et al., 2009). The uncertainty in particle count was taken as 3% (Kinney and Pui, 1991). Uncertainty in the wall loss rate constant (Equation 8), $k$, was taken as 5%, based on

computation of the average 95% confidence interval uncertainty of the fitted value of $k$ for each measurement set. The uncertainty in D was taken at 10%, while the resulting uncertainty in M was determined through a linear relation with D (Valencia and González, 2011), with $\Delta M = p \Delta D$ and $p = -3 \times 10^4 \, \text{g.s.mol}^{-1}\text{m}^{-2}$, with $p$ chosen to fit the values of M and D used in Saleh et al. (2013b) . The uncertainty in $\alpha$ and $C_{sat}$ for each experimental condition was computed as the standard deviation of 100 repeated simulations in which the above variables were randomly perturbed.

## 4 Results

### 4.1 SOA production

Descriptive statistics for the mean SOA distributions obtained under the various experimental conditions are given in Table 2. The SOA concentration in the reactor at low humidity was approximately $250 \, \mu\text{gm}^{-3}$ using either precursor, resulting in a 1:50 and a 1:100 dilution in the chamber at nominal and low loading conditions, respectively. For both precursors, increasing the

humidity in the reactor led to a significant increase in the total particle loading and a shift to smaller mean particle sizes,



especially in the case of engine SOA. Figure 6 shows typical size distributions of the generated SOA using α-pinene and engine exhaust at low and high humidity.

## 4.2 Evolution of $d_{cs}$

Figure 7 shows typical evolution of the condensation sink diameter versus time for α-pinene and engine SOA. Results are summarized in Table 3. For all conditions, $d_{cs}$ ceased to change within tens of minutes. In addition, we found that, for both precursors, the low loading conditions resulted in a greater extent of evaporation than the high loading conditions (high vs low: α-pinene SOA, $p<0.0006$; engine SOA, $p<0.07$). This effect is illustrated in Figure 8 for two typical engine SOA cases at the two loading conditions. For the baseline loading condition of 5 µgm$^{-3}$, these observations demonstrate that when equilibrium was reached, particles contained sufficient volatile material to induce observable particle shrinkage should it have evaporated.

## 4.3 Computed alpha and $C_{sat}$

As shown in Table 3, saturation concentrations ranged between 0.2-0.6 µgm$^{-3}$ and between 0.55-0.85 µgm$^{-3}$ for engine exhaust SOA under dry and humid conditions, respectively. For α-pinene SOA, $C_{sat}$ ranged between 0.35-0.65 µgm$^{-3}$ and between 0.65-1.2 µgm$^{-3}$ under dry and humid conditions, respectively. For both precursors, increasing the humidity in the reactor produced a modestly more volatile SOA (α-pinene SOA, $p<0.02$; engine SOA, $p<0.0008$).

For both α-pinene and engine SOA, the evaporation coefficient was on the order of 0.1. Differences in $\alpha$ between a-pinene and engine SOA at low humidity were not statistically significant. On the other hand, humidity was a significant factor. The experiments at higher humidity exhibited a lower evaporation coefficient, highly significant in the case of engine SOA (engine SOA, $p<0.0005$).

Within measurement uncertainty, the value of the wall contribution factor used in the simulations did not affect the values of the computed saturation concentration, the evaporation coefficient, or the goodness of fit of the model to the data. This is due to the fact that the particle loss to the walls during evaporation always represented a small fraction of the suspended particles (typically less than 15%). To simplify presentation, the results reported above assumed a value of unity $\omega$.

## 5. Discussion and Conclusions

Despite its importance to understanding and predicting regional air quality, the phase equilibration kinetics of engine-derived SOA has not been previously reported. In this study we investigated phase equilibration of SOA generated from engine exhaust under various conditions, at mass loadings typical of the urban atmosphere. We found that, like α-pinene SOA, these aerosols re-equilibrated within tens of minutes following a quasi-step change in saturation ratio. Key results are summarized in Table 3. Limitations of this study included the use of model precursors and simulated atmospheric photochemistry to generate SOA, that only one temperature was examined, and that the engine SOA were generated using a single-cylinder air-cooled engine





operating at one speed and load. Therefore, its emissions may not well reflect those generated by an urban vehicle fleet. Limitations notwithstanding, the SOAs studied here exhibited effective saturation concentrations on the order of $10^{-1}$ μgm$^{-3}$, similar to previous estimates for engine-exhaust and α-pinene SOA (Chen et al., 2013;Chhabra et al., 2015;Zhang et al., 2015;Tkacik et al., 2014), and also similar to those found in urban and rural environments (Paciga et al., 2016;Cappa and

5 Jimenez, 2010;Saha et al., 2017).

We found engine exhaust and α-pinene SOA evaporation coefficients to be of the order $10^{-1}$, consistent with values reported by Saleh et al. (2013b) for α-pinene SOA using a similar methodology to the current study. Our values were also consistent with reports from thermodenuder-based studies of dicarboxylic acids commonly found in atmospheric SOA and for concentrated ambient aerosols (Saleh et al. (2012). Similarly, Zhang et al. (2015) reported that SOA produced from the

10 oxidation of pure alkanes exhibited an effective evaporation coefficient of around 0.1. These values all suggest that ambient OA, whether anthropogenic or biogenic, should reach equilibrium phase-partitioning within minutes to hours (Saha and Grieshop, 2016;Saha et al., 2015;Saha et al., 2017). Krechmer et al. (2017) estimated an evaporation coefficient closer to 1 for compounds with $C_{sat}$ less than 10 μgm$^{-3}$. Taken together, these studies indicate that both anthropogenic and biogenic SOA are not intrinsically resistant to mass transfer, and that under most conditions their phase partitioning can be modeled assuming

thermodynamic equilibrium, considerably simplifying computational models of regional air pollution.

This conclusion is inconsistent with several previous studies that reported much lower values of the effective evaporation coefficient (Vaden et al., 2011;Grieshop et al., 2007;Cappa and Wilson, 2011), and which explained slow evaporation as an outcome of intra-particle mass transfer inhibition arising from the highly viscous or glassy amorphous state of SOA particles. These studies suggest that SOA remains chronically out of thermodynamic equilibrium, and, therefore, that equilibration

kinetics must be explicitly treated in regional air quality models. However, as pointed out in the introduction, these studies did not simultaneously determine the volatility of the aerosols under study; rather, they assumed volatility values from previous reports that may not have reflected the aerosols under study. As such, the observed slow or stalled evaporation reported in these studies and others (e.g. (Renbaum-Wolff et al., 2013;Perraud et al., 2012)) may have derived from unexpectedly low volatility rather than slow kinetics. In addition, these studies investigated particle evaporation under quasi vapor-free

conditions in which, no matter how much material had evaporated, the aerosol could not reach phase equilibrium, pushing the remaining particle material to ever lower volatility, and therefore ever slower evaporation.

While there have been several studies linking low relative humidity to slow equilibration kinetics (Song et al., 2015;Grayson et al., 2016;Saukko et al., 2012;Renbaum-Wolff et al., 2013), we found no significant difference in equilibration kinetics at <10% and 60% humidity, as characterized by the evaporation coefficient.

Regarding the larger question of whether SOA particles equilibrate rapidly in the environment, we found that, for typical ambient loadings, particles attained equilibrium within an hour, and that their evaporation behavior was well described by a model that lumped all resistance to mass transfer at the interface with the surrounding vapors, and which therefore neglected intraparticle diffusion resistance. We found this to be true regardless of the humidity level, the source of the SOA, or the loading.

The observed evaporation behavior was inconsistent with a picture in which volatile material is unable to escape due to intraparticle diffusion resistance, because when we used lower particle loading, the particles simply responded by evaporating more, and with roughly the same evaporation coefficient as the higher loading condition. In addition, when particles were maintained in a humid environment, the mass transfer resistance did not decrease relative to the dry condition. Had low particle

viscosity greatly inhibited mass transfer, we would expect particles in the humid condition to exhibit intrinsically faster kinetics (i.e. greater alpha).

Using an evaporation coefficient of 0.15, Figure 9 shows the dependence of the equilibration time ($\tau$) on both particle loading and particle diameter. As expected, increasing the particle diameter increases the equilibration time, since the condensation sink decreases, whereas increasing the particle loading decreases it. In all cases, $\tau$ is under 40 minutes for typical ambient

concentrations of organic aerosols (5 $\mu$gm$^{-3}$ < $C_{OA}$ <15 $\mu$gm$^{-3}$). This is far shorter than the timescales of chemical transformations in the atmosphere, which typically range between a few hours and several days (Seinfeld and Pandis, 2016). Thus, our estimates of the equilibration time in the atmosphere support the assumption of thermodynamic equilibrium commonly assumed in models.

## Acknowledgements

The authors acknowledge the assistance of Mr. Nareg Karaoghlanian and Mr. Ezzat Jaroudi in setting up the experimental apparatus and instrumentation and Mr. Joseph Nassif for fabricating the OFR.  The authors also thank Dr. Jose Jimenez for his advice on designing the OFR.

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



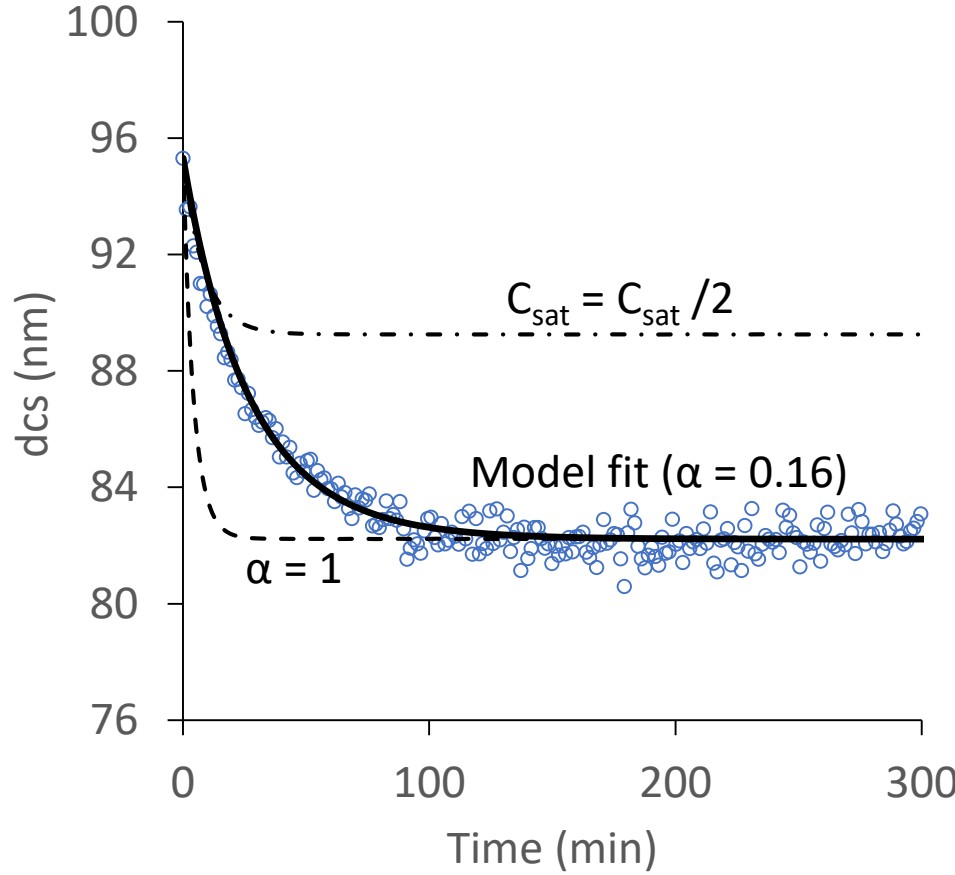

**Figure 1: The evolution of the condensation sink diameter after dilution and the model fit with α = 0.16. The figure also shows the model evaporation with α = 1, where a more rapid equilibration occurs, and Csat = 0.5 × Csat, where equilibration is achieved with a smaller change in d$_{cs}$.**




**Figure 2. Schematic of experimental setup.**

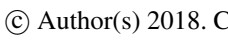



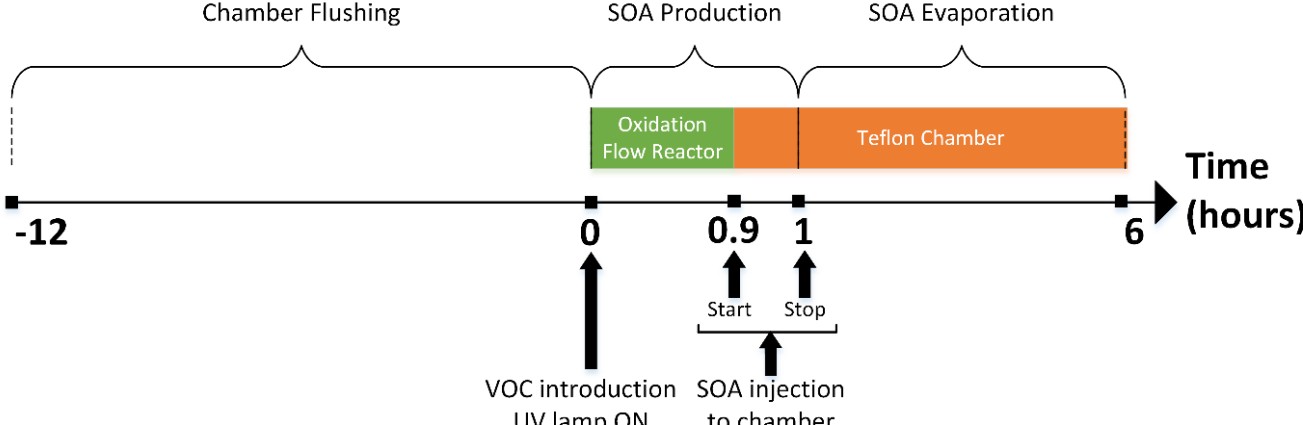

Figure 3. Experimental timeline. Preceding all experiments, the chamber was flushed with zero air for12 hours. Time 0 marks the start of the experimental procedure, with the introduction of the volatile precursors into the OFR and turning on the UV lamps. After the particle concentration in the OFR reaches steady state, the OFR outlet is diverted into the Teflon chamber until the mass concentration reaches the desired value. The size distribution in the chamber is monitored for up to 300 minutes. Colored bands in the chart indicate what the SMPS is sampling.



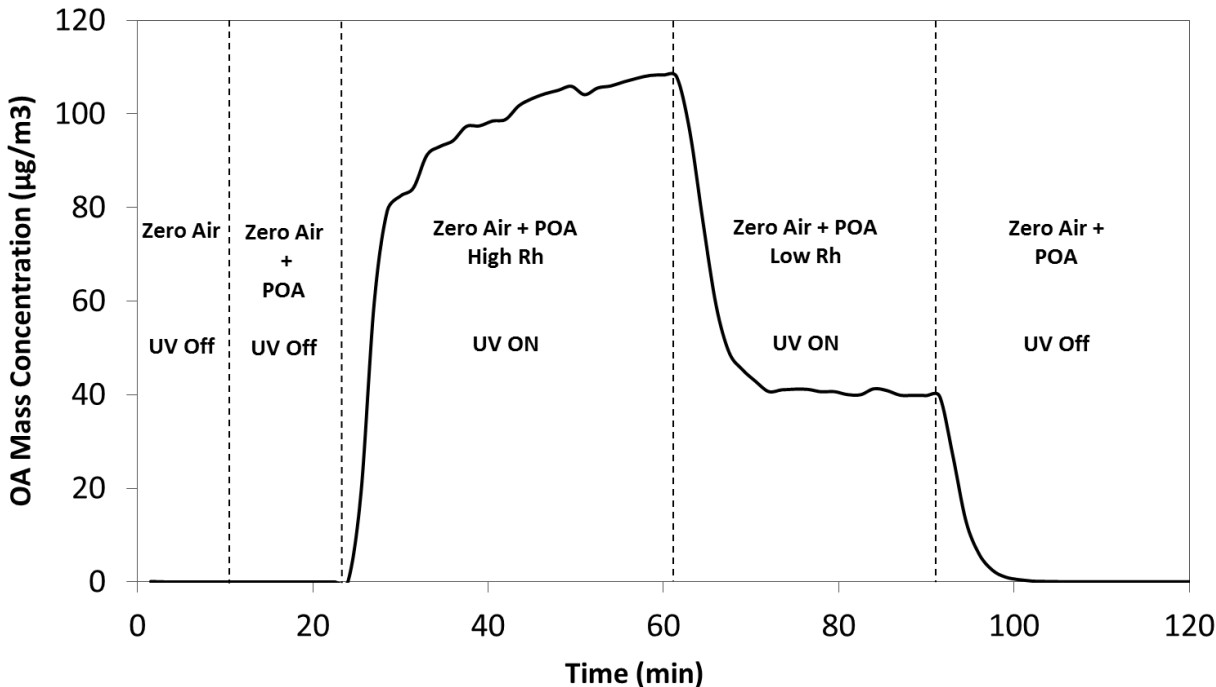

**Figure 4.** The mass concentration evolution during preliminary measurements of the production of SOA in the oxidation reactor as a function of time and variables (UV: ultraviolet radiation)







**Figure 5. The raw and wall deposition-corrected $d_{cs}$ for a typical experiment. After an initial period in which evaporation dominates, a steady rise in $d_{cs}$ due to wall deposition takes over.**


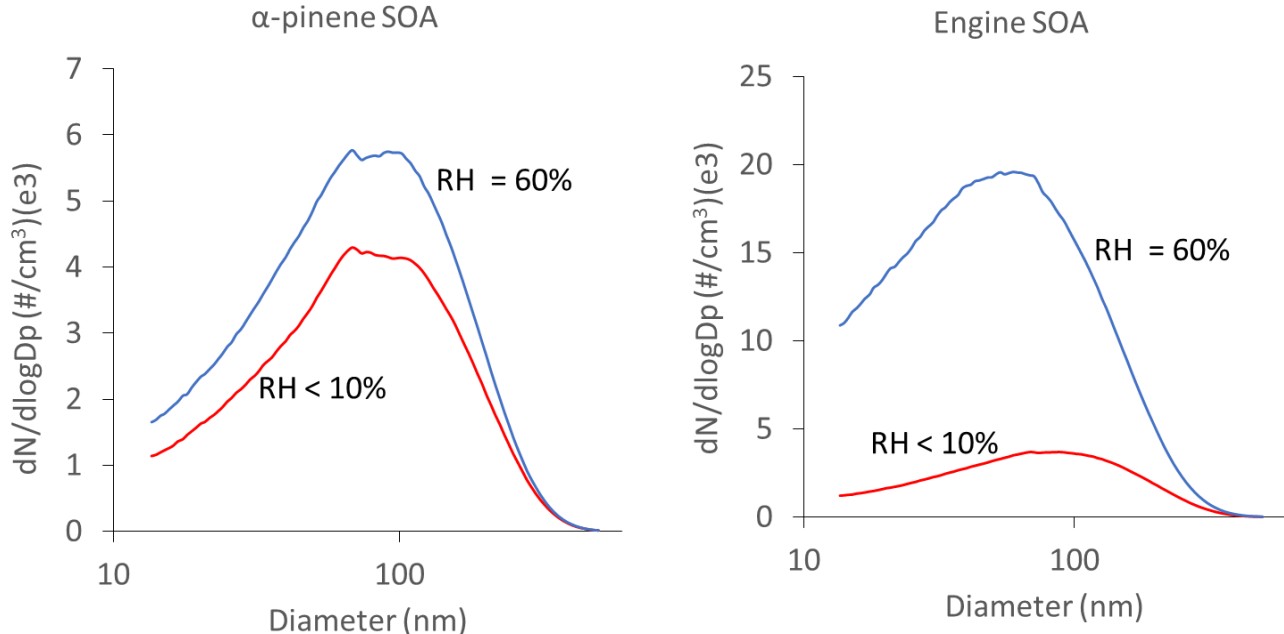

**Figure 6. Particle size distribution of α-pinene SOA and engine SOA inside OFR under low and high humidity.**



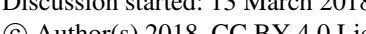

Figure 7. Sample experimental results and model fit of condensation sink diameter vs time for engine and α-pinene SOA under dry and humid conditions.





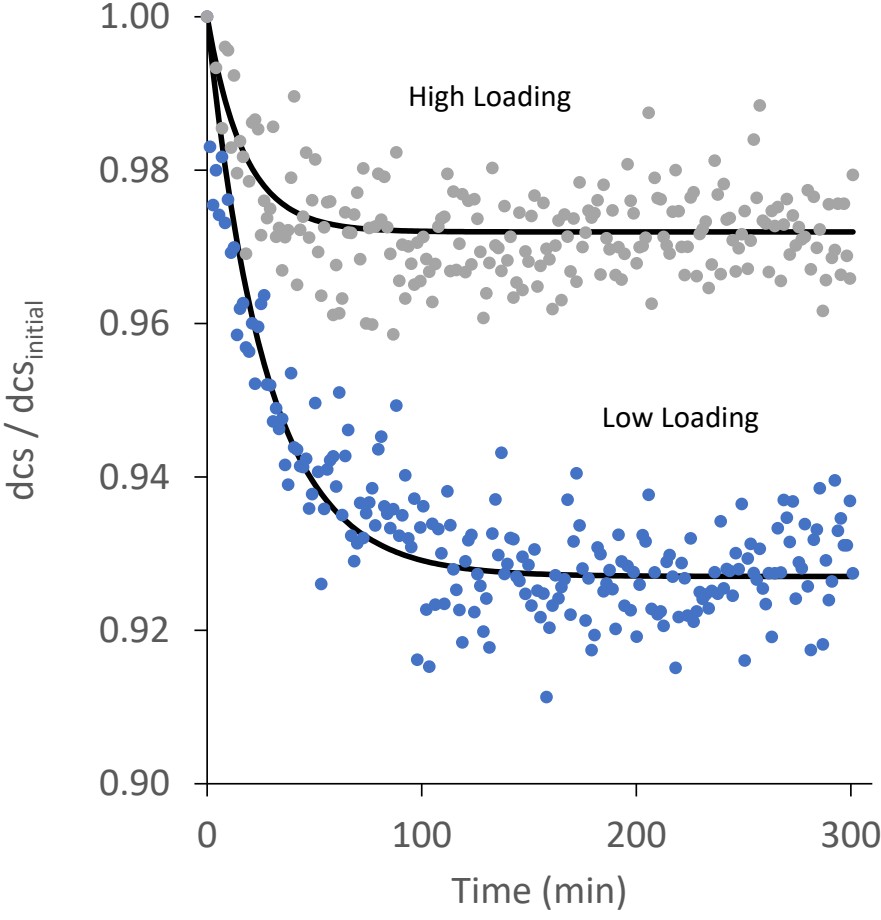

**Figure 8. Ratio of dcs over time to the value of dcs upon first dilution in the chamber at low and high mass loadings. At lower loading, the change in dcs is larger, indicating that particles surfaces are not depleted of volatile material.**




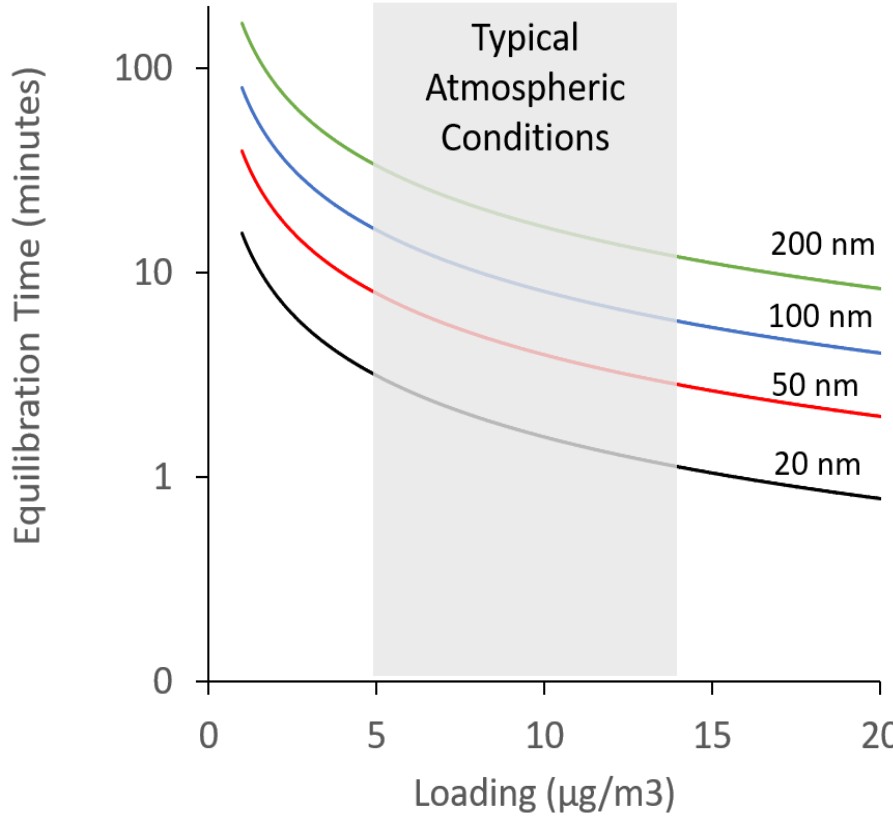

**Figure 9. Equilibration time scales for aerosol systems of different diameters and particle loadings using α = 0.15.**



**Table 1. Experimental conditions**

| Precursor | α-pinene | | | Engine Exhaust | | |
|---|---|---|---|---|---|---|
| Number of measurements | 5 | 5 | 5 | 6 | 6 | 5 |
| Rel. Humidity | < 10% | < 10% | 60% | < 10% | < 10% | 60% |
| **Evaporation chamber** | | | | | | |
| Nom. initial $C_{OA}$ (µg/m$^3$) | 5 | 2.5 | 5 | 5 | 2.5 | 5 |
| Temperature (K) | 305 | 305 | 305 | 305 | 305 | 305 |
| Residence time (min) | 300 | 300 | 300 | 300 | 300 | 300 |
| **Oxidation flow reactor** | | | | | | |
| Dry air flow (lpm) | 20.7 | 20.7 | 20.7 | 17 | 20.7 | 16 |
| Humidifier flow (lpm) | 4.0 (dry) | 4.0 (dry) | 4.0 (wet) | 3.0 (dry) | 4.0 (dry) | 4.0 (wet) |
| Precursor flow (lpm) | 0.30 | 0.30 | 0.30 | 5.00 | 0.30 | 5.00 |



**Table 2. Size distribution of SOA in the reactor. The results at 10% RH are the means of 5 measurements using α-pinene and 6 measurements using engine exhaust. The results at 60% RH are the means of 5 measurements using either precursor. Reported as mean ± SD.**

| | Precursor | | | | | |
|---|---|---|---|---|---|---|
| | α-pinene | | | Engine Exhaust | | |
| Rel. humidity | < 10% | < 10% | 60% | < 10% | < 10% | 60% |
| Nom. chamber $C_{OA}$ (µgm⁻³) | 5 | 5 | 5 | 5 | 2.5 | 5 |
| Reactor $C_{OA}$ (µgm⁻³) | 258 ± 47 | 245 ± 52 | 416 ± 39 | 255 ± 15 | 249 ± 19 | 739 ± 178 |
| Number Concentration (10⁵ cm⁻³) | 2. 1 ± 0.3 | 2.2 ± 0.1 | 4.1 ± 0.6 | 2.4 ± 0.4 | 2.7 ± 0.4 | 15.7 ± 4.8 |
| $D_{Mean}$ (nm) | 88 ± 4 | 90 ± 5 | 84 ± 3 | 86 ± 4 | 83 ± 5 | 63 ± 2 |
| $D_{Mode}$ (nm) | 90 ± 7 | 87 ± 7 | 82 ± 5 | 84 ± 4 | 78 ± 4 | 51 ± 6 |
| Geometric standard deviation | 2.1 ± 0 | 2.1 ± 0 | 2.1 ± 0 | 2.1 ± 0 | 2.1 ± 0 | 2.1 ± 0 |



**Table 3. Mean(± SD) conditions and results of chamber experiments under all conditions. Csat and evaporation coefficient SD based on perturbation analysis (see text).**

| | Precursor | | | | | |
|---|---|---|---|---|---|---|
| | α-pinene | | | Engine Exhaust | | |
| Humidity | < 10% | < 10% | 60% | < 10% | < 10% | 60% |
| $C_{OA}$ ($\mu$gm$^{-3}$) | 5.2 ± 0.6 | 2.5 ± 0.3 | 4.9 ± 0.5 | 5.5 ± 0.3 | 2.6 ± 0.3 | 5.3 ± 0.6 |
| Number concentration ($10^3$ cm$^{-3}$) | 5.0 ± 0.4 | 2.7 ± 0.30 | 5.5 ± 0.2 | 5.4 ± 0.8 | 2.9 ± 0.4 | 13.8 ± 0.8 |
| Initial $d_{cs}$ (nm) | 103.1 ± 6.0 | 99.0 ± 4.0 | 98.8 ± 2.9 | 101.1 ± 4.2 | 98.0 ± 4.9 | 72.8 ± 2.3 |
| $d_{cs}$ change (nm) | 6.5 ± 0.3 | 8.5 ± 0.7 | 10.5 ± 1.8 | 4.5 ± 1.3 | 6.8 ± 2.1 | 6.3 ± 1.7 |
| Csat ($\mu$gm$^{-3}$) | 0.64 ± 0.05 | 0.40 ± 0.04 | 0.93 ± 0.16 | 0.44 ± 0.09 | 0.33 ± 0.08 | 0.81 ± 0.13 |
| Evaporation coefficient | 0.11 ± 0.02 | 0.15 ± 0.03 | 0.10 ± 0.04 | 0.13 ± 0.02 | 0.18 ± 0.04 | 0.06 ± 0.01 |