# Peer review of "Figure S1. OA Mass concentration and Ozone concentration evolution during SOA production during preliminary measurements."

_Atmospheric Chemistry and Physics, 2018_

## Referee Comment (RC1) · Anonymous Referee #1 · 12 Apr 2018

Review of "Is mass transfer in secondary organic aerosol particles intrinsically slow? Equilibration timescales of engine exhaust and a-pinene soa under dry and humid conditions" by Atwi et al.

The authors present observations of the extent of evaporation for SOA formed either from alpha-pinene photooxidation or from exhaust from a small engine, after dilution from relatively high concentrations (100's of micrograms/m3) to relatively low concentrations ($\sim$5 micrograms/m3). The observe the particles to shrink by a small amount, typically $\sim$5-10%, to a new, relatively constant value on a timescale of $\sim$1 h. The

experiments appear to be of good quality.

They interpret these observations in terms of a single-component evaporation model from which they deduce the effective saturation concentration and evaporation coefficient. I have substantial concerns regarding the interpretation. There is a fundamental limitation from the outset, namely that the authors assume that the particles are made of one compound and thus the evaporation can be characterized through a single value of the effective saturation concentration and evaporation coefficient. It is now well-established that SOA is composed of many, many compounds, not one compound, and that treatments of SOA as if made of one component are too simple and can yield results that are not linked to true physical properties. (Think of the two-product model, for example.) This fundamental assumption dictates the entire analysis and the results and inferences that can possibly result. The team has previously published results using similar methods and with the same core assumption. I believe that this work should not be published because it perpetuates what I believe to be an overly simplified analysis framework. I strongly encourage the authors to expand the scope of their model to treat the SOA in a more physical manner, at minimum as if composed of a distribution of compounds with varying volatilities. This would not get at issues of oligomerization (mentioned by the authors as potentially important) and how this might impact their interpretation. But it would at least move it in the right direction in terms of using a more realistic representation of the physical properties. That said, I suspect that the authors will quickly find that if they expand their modeling framework they will end up with many unknown model parameters and many degenerate solutions, greatly complicating interpretation. I wish I could see a more favorable outcome for this work, but unless the core assumption underlying the analysis is changed I do not see a path to publication of this work.

Below I provide specific comments on the manuscript, and further discussion of my core concern regarding the analysis/interpretation framework.

Abstract: it is by now well-established that SOA comprises a variety of compounds with

a distribution of vapor pressures. Given this, it is now unclear what it means to have an "effective thermodynamic saturation concentration" of a single value.

P2/L17: There are many models that use a kinetic approach to partitioning. The complexity of the process representation may be limited, but kinetic partitioning is nonetheless used in numerous atmospheric models. "Some" models assume instantaneous equilibrium.

P3/L13: The effective evaporation coefficients determined by Saleh/Saha et al. are not necessarily the same as those reported in the previous paragraph. The determination and reporting of an evaporation coefficient is highly dependent upon assumptions made in the interpretation. There is a direct link between the assumed volatility distribution and the obtained evaporation coefficient. The comparison here does not recognize sufficiently this complexity. This is noted two paragraphs later, but the nuance is lost in this paragraph. I suggest bringing these together to minimize misunderstanding.

P3/L25: better as "estimated by traditional interpretation of smog chamber experiments."

P3/L26: Even non "aged" SOA may have oligomers.

P3/L31: When the components that are very low volatility, and thus evaporate only a little bit upon e.g. dilution, the evaporation experiments mentioned are not able to establish the evaporation coefficients of these components. A noticeable change in particle size/mass must be observed for an evaporation coefficient to be determined. Thus, the cited studies determined evaporation coefficients for the components that contributed most to the mass change. This issue needs to be recognized throughout the manuscript. The authors can only determine evaporation coefficients for components that evaporate non-negligibly and that comprise a notable fraction of the volume.

P3/L33: There are a multitude of studies that have studied SOA from anthropogenic precursors. I do not understand what the authors mean here in saying that lab studies

focus exclusively on biogenic SOA. This is simply incorrect.

P4/L27: An explicit assumption of the model is that the system can be modeled using a single vapor pressure. As noted above, it is well established that SOA is a distribution of species of differing volatility. Thus, the single-volatility approximation used here is limited in terms of its physical realism. While it does not "require a priori knowledge about aerosol volatility," it does a priori assume that SOA is describable by a single volatility. This substantial limitation should be noted. Moreover, the limitations of this fundamental assumption should be assessed. What if the authors assumed that the particles were composed of two compounds with volatilities that differed by one or-der of magnitude and existed in proportions that allowed for the average volume loss observed? Or of one higher volatility and one non-volatile component? Would the time-dependent trajectory (i.e. equilibration) differ from the single-component model, or is the trajectory completely independent of the assumption of a single-component system? This can be theoretically explored by the authors via minimal modification of their modeling framework. It will, however, introduce further unknowns. Thus, they might consider a systematic exploration of the assumption. To me, this foundational model assumption is the fundamental limitation of this entire study.

P6/L6: It is stated that Csat and alpha are the "unknowns" since the other parameters can be estimated. But it is unclear how the wall term omega is known. Bounds can be established (zero to one), but it is not known. This should be discussed.

P7/L16: mlpm is non-standard.

Fig. 2: I find the schematic unclear. It appears that 25 lpm is going in to the chamber but only 1 lpm is coming out. Is this correct? The authors also use a lot of symbols in the figure that could benefit from a legend. And some flows are labeled, others are not. Consistency in labeling would be helpful.

Fig. 3: What is "POA" in this figure? This typically means "primary organic aerosol." How is the OA concentration zero when there is POA around?

Fig. 3: Is there no black carbon for the engine emissions? The authors seem to assume that all mass is OA. Has this been confirmed?

Fig. 6: Would surface area distributions be more appropriate, since the condensational sink is what matters?

Fig. 6 vs. Fig. 7: I do not understand the relationship between these figures. The peak in the number distribution for, for example, the dry Engine SOA case, is near 100 nm. So how is the diameter for the condensation sink only 67 nm?

Fig. 8 and P10/L6: The nature of the "low" versus "high" loading conditions is unclear. This is not discussed in the methods. How were these conditions obtained? Are these just random differences between experiments? Or some systematic examination of loading? And what is "high" and what is "low." This doesn't appear to be stated clearly. I find this discussion unclear, including the extrapolation to the particles having sufficient volatile material had it evaporated.

P10/L17: The statement that "humidity was a significant factor" is sufficiently imprecise to be incorrect. This was only found to be the case for one of the two systems considered. So for one it is not a factor, the other it is.

P11/L16: The authors again frame these literature studies in contrast to the current study as fundamentally different. They simply, as the authors note, build from different assumptions but are, as least in theory, reconcilable. This is already known, and even mentioned by some of the studies cited (and some that are not cited). It is suggested that the authors reframe this to indicate that their study is further support for the idea that the evaporation coefficient cannot be determined without clear understanding of the underlying volatility distribution. Also, at the end of this paragraph the authors appear to link lower volatility to slower evaporation, which is fine, but in the context of evaporation coefficients this is inconsequential.

P11/L30: The authors observed that the particles stopped shrinking (after wall correc-

tion) in a noticeable way after ∼100 mins up to a total time of 300 mins. From this they are concluding that the particles have reached equilibrium. How do they know? What if there is a fraction of the particle that evaporates rapidly and a fraction that evaporates/equilibrates slowly? Much slower than the timeframe experimentally covered here? How can the authors rule out the possibility that, had they looked over even longer timescales they would not have observed a continued evolution of the system? This comes back to the fundamental assumption that the authors interpret everything within a single-component framework. I believe that the authors need to push beyond this single-component framework to consider the implications, at the minimum, of what would happen if they took a fuller approach.

P12/L2: I find this a bit unconvincing for the following reason. The authors have not done a sufficient job of examining the loading dependence. The have examined exactly two concentrations that differ by only a factor of two, not performed a systematic exploration. And, in the example shown in Fig. 8, this is not representative of the average results (deduced from table 3), which show much smaller differences, on average, between the two loading cases. Further, as with the entire study, this interpretation is very much limited by the fundamental assumption of a single-component aerosol.

P12/L10: I find the statement regarding timescales of chemical transformations unclear. Chemical transformations occur on time scales ranging from seconds (or much, much less) to much longer. Chemical transformations, like partitioning, occur on a continuum of timescales. The authors cite Seinfeld and Pandis here (which is missing from the reference list). I ask them to point to the page where this range of timescales is given for "chemical transformations."

---

## Referee Comment (RC2) · Anonymous Referee #2 · 18 Apr 2018

The authors present the evaporation kinetics of SOA formed from engine exhaust and $\alpha$-pinene SOA formed via photooxidation using their homemade oxidation flow reactor (OFR). The SOA produced by the OFR has mass loadings in excess of 100 $\mu$g m-3, and 2.5 $\mu$g m-3 and 5.0 $\mu$g m-3 of the SOA is injected into their Teflon chamber for evaporation measurements. During the course of the evaporation measurements there is a small change in the peak diameter corresponding to 5-10% change in the volume of the particles after about an hour. When the size distribution is 'stable' the authors propose that there is no subsequent change in particle size because the SOA has

reached equilibrium with the vapor phase. It is important to note, the peak diameter increases at long times due to the size-dependent wall loss exhibited during control experiments using ammonium sulfate aerosol. The evaporation kinetics are modeled with what can only be described as a simplistic single product model.

Beyond this I have major concerns about one of the main underlying assumptions that is used in the interpretation of the results. For the particles in the Teflon chamber they are assumed to evaporate or be lost to the walls and the authors treat these particles as able to evaporate or not based on an omega parameter, and this treatment is fine. However, the fate of the gas phase vapor that evaporates in the chamber is assumed to only stay in the gas phase and not partition or be lost to the walls. Recent studies have shown that gas vapors with a saturation vapor pressure between $0.01 - 100$ $\mu$g m-3 will be lost to the walls, which act as a sink for all semivolatile vapors.[1-5] If it is assumed the saturation vapor concentration put forth in the paper is true, then it would be expected that the vapor would be lost to the walls during the course of the evaporation measurement and the system would never truly be at equilibrium. Without a measure of gas phase vapor concentrations or something continually scavenging the gas phase vapor (i.e. activated charcoal) simply letting the Teflon chamber act as a vapor sink makes it impossible to constrain the set of equations used by the authors. Because of this and the simplistic model used, mentioned at length in the other review, I do not recommend the publication of this paper.

Comments:

Pg 3 ln 31: It seems odd cite those that did while not citing those that did not.

Pg 3 ln 33: There have been many studies that have investigated anthropogenic SOA... perhaps the authors mean that volatility measurements of anthropogenic SOA are what is limited.

Pg 4 ln 18: I find the discussion here needs to be significantly changed. Pg 4 ln 18 references the evaporation coefficient $\alpha$, but it is not present in the set of equations that

have been presented to that point.

Pg 4 ln 25: What benefit and impact to the broader community is there to using such a simple model?

Pg 5: The significant limitation of this work is underlying assumption that the vapor phase only interacts with the particles in the chamber. As discussed above, not taking vapor wall losses into account makes this work untenable.

Pg 6: In this work the humidity of the evaporation chamber and the OFR are the same, to probe the effect of viscosity it would be more appropriate to try to study dry (formation) vs. wet (evaporation) and vice versa as performed in Wilson et al.[6] This comparison is important because the composition of SOA can be heavily dependent on the RH of formation, for an example see Hinks et al. for Toluene SOA.[7] As a result, the differences shown here between ~10% and 60% RH may not be the appropriate comparison.

Pg 7 ln 5-13: What are the dominant VOC emissions from this engine? It is difficult to conclusively say anything about the volatility of the SOA produced from this source without some knowledge of the precursors VOCs that are emitted. Also, are the primary emissions from the engine filtered out with the HEPA filter? If the primary organic aerosol is not filtered then what is the mass loading prior to the lights being turned on? Figure 4 suggests there is no OA mass concentration, but doesn't explicitly mention how this is measured or what experiment this is from.

Pg 7/8: Another point about the experimental setup is the difference in temperature between the OFR (24 C) and the evaporation chamber (32 C). Was there any reason for not operating both chambers at the same temperature?

Pg 9 ln 4: Does k change for every experiment? What range of k values are present for the experiments? In the abstract the saturation concentration is inconsistent with the results presented in the paper (0.02 – 0.11 vs. 0.2 – 1.2 (see pg 10)). Also in the

abstract there is a mention to the enthalpy of vaporization (150 Kj/mol) and I don't see this referred to in the rest of the paper. If this it isn't mentioned in the paper why is it in the abstract?

(1) Krechmer, J. E.; Pagonis, D.; Ziemann, P. J.; Jimenez, J. L. Environmental Science & Technology 2016, 50, 5757, DOI:10.1021/acs.est.6b00606 (2) Krechmer, J. E.; Day, D. A.; Ziemann, P. J.; Jimenez, J. L. Environmental Science & Technology 2017, 51, 11867, DOI:10.1021/acs.est.7b02144 (3) Huang, Y.; Zhao, R.; Charan, S. M.; Kenseth, C. M.; Zhang, X.; Seinfeld, J. H. Environmental Science & Technology 2018, 52, 2134, DOI:10.1021/acs.est.7b05575 (4) La, Y. S.; Camredon, M.; Ziemann, P. J.; Valorso, R.; Matsunaga, A.; Lannuque, V.; Lee-Taylor, J.; Hodzic, A.; Madronich, S.; Aumont, B. Atmos. Chem. Phys. 2016, 16, 1417, DOI:10.5194/acp-16-1417-2016 (5) Trump, E. R.; Epstein, S. A.; Riipinen, I.; Donahue, N. M. Aerosol Science and Technology 2016, 50, 1180, DOI:10.1080/02786826.2016.1232858 (6) Wilson, J.; Imre, D.; Beránek, J.; Shrivastava, M.; Zelenyuk, A. Environmental Science & Technology 2015, 49, 243, DOI:10.1021/es505331d (7) Hinks, M. L.; Montoya-Aguilera, J.; Ellison, L.; Lin, P.; Laskin, A.; Laskin, J.; Shiraiwa, M.; Dabdub, D.; Nizkorodov, S. A. Atmos. Chem. Phys. 2018, 18, 1643, DOI:10.5194/acp-18-1643-2018